# The last-born daughter cell contributes to division orientation of *Drosophila* larval neuroblasts

Nicolas Loyer[1] & Jens Januschke [1]

Controlling the orientation of cell division is important in the context of cell fate choices and tissue morphogenesis. However, the mechanisms providing the required positional information remain incompletely understood. Here we use stem cells of the *Drosophila* larval brain that stably maintain their axis of polarity and division between cell cycles to identify cues that orient cell division. Using live cell imaging of cultured brains, laser ablation and genetics, we reveal that division axis maintenance relies on their last-born daughter cell. We propose that, in addition to known intrinsic cues, stem cells in the developing fly brain are polarized by an extrinsic signal. We further find that division axis maintenance allows neuroblasts to maximize their contact area with glial cells known to provide protective and proliferative signals to neuroblasts.

[1] Cell & Developmental Biology, School of Life Sciences, University of Dundee, MSI/WTB3 Complex, Dow Street, Dundee DD1 5EH, UK. Correspondence and requests for materials should be addressed to J.J. (email: j.januschke@dundee.ac.uk)

The orientation of cell division is important for cell fate choices and impacts on the morphology and function of tissues[1,2]. Therefore, perhaps not surprising, defects in spindle orientation have been linked to developmental defects and diseases[3]. Coupling spindle orientation to unequal segregation of fate determinants is one strategy during asymmetric cell division to generate different cell fates[4,5]. Spindle orientation also affects the placement of daughter cells after division. This can alter function and fate of the resulting daughter cells as the microenvironment in different positions can cause daughter cells to experience different signals[6].

In cells, an evolutionary conserved molecular machinery helps to position the spindle by anchoring the astral microtubules to cortical attachment sites[7,8]. A key challenge in this context is understanding the spatial information that determines the position of these attachment sites. The machinery anchoring microtubules at the cortex frequently depends on the axis of polarity of the dividing cell. In those contexts, the symmetry breaking event that polarizes a cell and gives the polarity axis its orientation also determines the orientation of the subsequent division. Microtubules can act in many contexts such as a signal biasing with which orientation cells polarize (reviewed in ref. [9]), but a variety of other polarizing cues exist that polarize cells and orient their division.

Embryonic neural stem cells (neuroblasts (NBs)) in *Drosophila* for instance can use spindle microtubules to deliver components of the microtubule anchoring machinery to the cortex[10]. These cells can also read extrinsic cues, orienting their division perpendicular to the overlying epithelium[11]. This is mediated by G-protein coupled receptor signalling recruiting factors directly orienting the spindle towards this signal[12]. In other contexts, E-cadherin (E-Cad) rich cell–cell adhesion sites provide spatial information to orient the mitotic spindle[13–15]. In the case of *Caenorhabditis elegans*, the site of sperm entry defines anterior–posterior polarity and the orientation of the first division[16]. The midbody resulting from this division is further used as a spatial cue orienting the subsequent P1 cell division[17]. Cytokinesis is also linked to division orientation control in budding yeast, where the orientation of future divisions is biased by a landmark at the site of abscission[18]. Therefore, positional landmarks linked to cytokinesis can control the orientation of cell division.

An ideal system to study mechanisms that orient cell division are the highly proliferative NBs of the *Drosophila* larva that divide over many cell cycles with very little deviation in the orientation of division between different cycles. The mechanisms controlling this process are only partially understood. In NBs, cortical polarity is established by the activity of the Par complex[19–23]. The Pins (*Drosophila* homologue of LGN) complex[24–28] then couples the orientation of the mitotic spindle with apico-basal polarity, such that both are aligned. Interestingly, after each division the polarized localization of both complexes on the NB cortex is lost but reforms with the same orientation in the next mitosis[29,30]. Contrary to embryonic NBs[11], this occurs regardless of whether larval NBs reside within the brain or are in isolation in primary culture[31,32]. Currently, this process is believed to occur through the apically localized centrosome and microtubules, which act as cell intrinsic polarizing cues[33]. However, disruption of these cues, either through depolymerization of microtubules or mutation in *sas4* leading to loss of centrioles[34], only results in a partial defect of division orientation maintenance[32]. This suggests that other polarizing cues contribute in parallel to maintain the orientation of the axis of NB division.

Given that cytokinesis-related cues can direct spindle orientation in other cell types, we hypothesized that NBs could use a spatial cue provided by their last-born daughter cell to orient cell division in the subsequent mitosis. Indeed, we found that NBs align their divisions with the position of the last-born daughter cell (called ganglion mother cell (GMC)). Disruption of the integrity of the NB/GMC interface, either through laser ablation or by depletion of proteins specifically localizing to this interface, including the midbody and midbody-associated structures, perturbs NB division orientation memory while it does not affect alignment of the mitotic spindle with cortical polarity. Thus we propose that the last-born GMC is an extrinsic polarizing cue for larval brain NBs in *Drosophila* orienting their axis of polarity and consequently division. Finally, our results suggest a physiological function for division axis maintenance in this context: preventing NBs from generating daughter cells between themselves and surrounding cortex glia maximizes NB/cortex glia contact surface.

## Results

**The division axis of NBs follows GMC movements.** To test our hypothesis that the orientation of NB division is under the influence of extrinsic cues provided by their daughter cells, we analysed the relationship of the orientation of NB division with the position of the last-born daughter cell (GMC). We used confocal imaging of whole mount brains to capture the entire volume of NBs over several rounds of divisions (Supplementary Movie 1) and developed a method to measure the deviation of their division axis in three dimension (3D) using the characteristic shape of telophase NBs as reference (Fig. 1a and Supplementary Fig. 1a). Using this method, we reproduced the previous observation[32] that maintenance of the division axis of NBs is affected, but not abolished, in *sas4* mutants (Supplementary Fig. 1b-d). This partial maintenance of the division axis upon loss of the *sas4*-dependent polarity cue is consistent with the possible continued activity of an additional spatial cue.

As previously reported[32], the division axis of control NBs is not perfectly maintained between cell cycles (angle $\alpha$, Fig. 1b, c). We next reasoned that, if the GMC provides a cue maintaining the division axis, the division axis of NBs should align with the position of the GMC when NBs polarize, i.e. when they start rounding at the onset of mitosis (Supplementary Fig. 1e, Supplementary Movie 2). Thus we tracked the position of the last-born GMC until NBs started rounding up, at which point we defined a NB–GMC axis (Fig. 1b, magenta arrow). Comparing this axis to the following division axis (Fig. 1b, green arrow) revealed that the position of the last-born GMC at the onset of NB rounding predicted significantly better the orientation of the subsequent division than the previous axis of division (angle $\beta$, Fig. 1c).

We further observed that, in cases for which the division axis was most highly misaligned with the previous one, the last-born GMC moved away from its initial birthplace, and the NB from which it derived re-aligned its subsequent division with the new position of that GMC (Fig. 1a). Importantly, this displacement of the GMC and the subsequent realignment of the division axis was not caused by a rotation of the entire brain nor of the NB-progeny cluster as only the last-born GMC and not its neighbouring cells significantly changed position (Supplementary Fig. 1f-h). Whether such GMC movements are physiologically relevant or experimentally induced is unclear. Nonetheless, our observation that NB divisions realign with displaced GMCs is consistent with the idea that the GMC is a spatial cue that orients NB divisions.

**Last-born GMC ablation affects NB division orientation.** If the last-born GMC controls NB division orientation, ablation should affect the maintenance of the axis of NB division (measured above as "angle $\alpha$"). We tested this idea by observing a NB

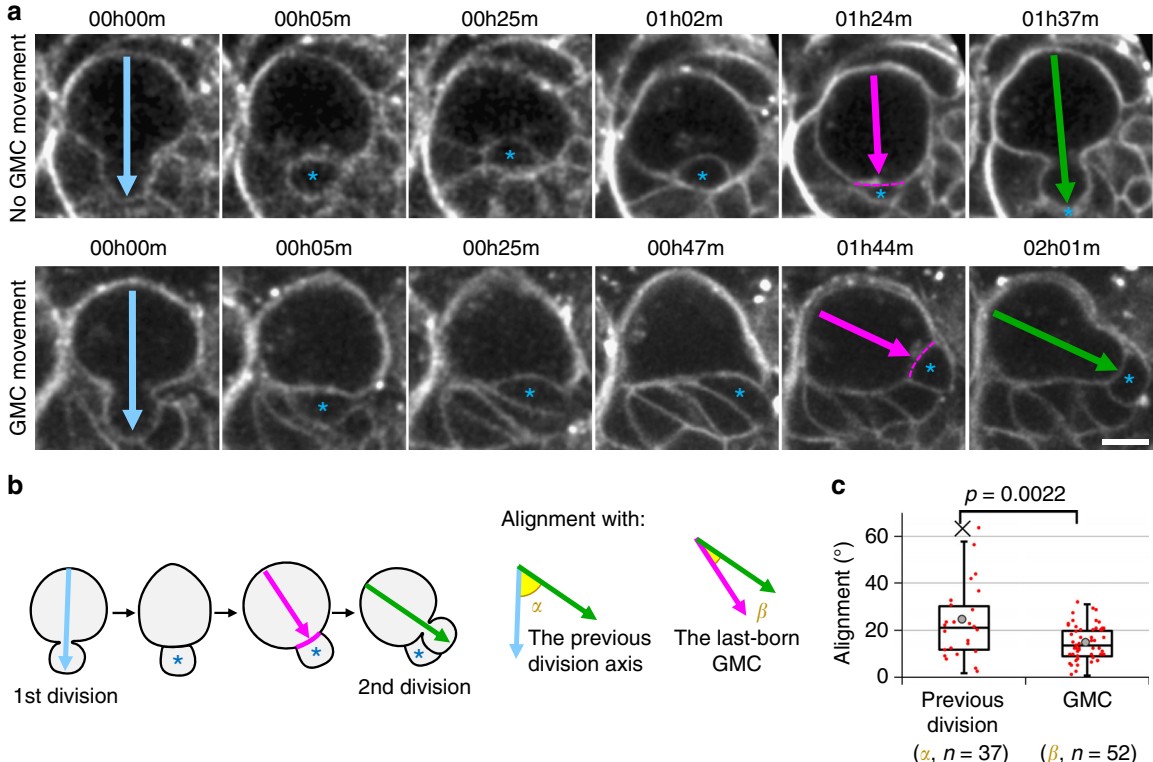

**Fig. 1** NBs align their division axis with GMC position. **a** Two successive divisions of a larval NB expressing the membrane marker PH::RFP. Top row: no obvious movement of the last-born GMC is observed following the first division; bottom row: last-born GMC movement. Arrows: division axis. Asterisk: last-born GMC. Dashed line: NB/GMC interface at prophase. Scale bar: 5 μm. **b** Angles measured in the previous panel and quantified in the next panel. Angle α corresponds to the alignment of the second division (green arrow) with the previous one (blue arrow). Angle β corresponds to the alignment of the second division (green arrow) with an axis (magenta arrow) defined by the GMC (blue asterisk)/NB interface (magenta line) at prophase. **c** Distribution of the angles described in the previous panel: alignment of the second division axis with the previous division axis (angle α, 24 ± 15° (standard deviation), $n = 37$) and with the GMC (angle β, 14 ± 7°, $n = 52$) from one experiment In this boxplot and every following one: cross: outlier; grey circle: average; red dots: individual measurements; centre line, median; box limits, upper and lower quartiles; whiskers, 1.5× interquartile range. $P$ values displayed over boxplots were calculated using a non-parametric two-tailed Mann–Whitney $U$ Test in this figure and all subsequent ones

division and destroying the GMC born from this division by biphotonic laser-mediated ablation. We then observed the following division and measured its alignment with the previous one. NBs immediately deformed toward the destroyed GMC following its ablation (Supplementary Fig. 2, Supplementary Movie 3). Therefore, we were concerned that this deformation and cellular debris generated close to the NB together with damage caused by the laser could affect the maintenance of the division axis and bias our analysis. To account for such effects, we performed "control ablations" by ablating other cells in contact with the NB, away from the last-born GMC (Fig. 2a, b).

Control ablations also resulted in NB deformation towards the destroyed cell (Supplementary Fig. 2, Supplementary Movie 4). However, they did not affect the maintenance of NB division orientation. In contrast, ablation of the last-born GMC significantly affected this process (Fig. 2c, Supplementary Movie 5). Importantly, NBs that misoriented their division following ablation of the last-born GMC and then divided again aligned this third division with the previous, misoriented division (Fig. 2d–f). Thus ablation of the last-born GMC results in a transient defect in the orientation of the NB division axis that is restored upon the generation of a new GMC. This is further consistent with our (Fig. 1) and a previous observation[32]. Finally, we investigated whether older GMCs also participate in division axis maintenance by targeting the GMC generated one cell cycle earlier rather than the last-born GMC. Ablation of the older GMC led to a small but non-significant increase of the division

axis deviation compared to control cuts (targeting a cell away from the last-born GMC, Fig. 2c), which might be attributed to indirectly affecting the last-born GMC via generation of cellular debris, direct damages by the laser and deformation of neighbouring cells toward the ablated cell. These results suggest a prominent role for the last-born GMC over the entire GMC cluster produced.

**Last-born GMC ablation affects NB polarity axis orientation.** In NBs, the spindle is aligned with the apico-basal polarity axis. Altered division orientation caused by GMC ablation could therefore be the result of misalignment of the mitotic spindle with the apico-basal polarity axis. Alternatively, the orientation of the polarity axis itself could be affected. We tested this by repeating GMC ablation experiments in NBs expressing the apical polarity marker Baz::GFP, together with the centriole marker Asl::YFP to visualize the mitotic spindle poles. Following GMC ablation, every case of division axis misalignment ($n = 7$ cases with deviation >45°) displayed a misplaced apical crescent with which the spindle properly aligned (Fig. 3a, b). Thus defective division axis maintenance upon GMC ablation results from altered orientation of the polarity axis and not from defects in downstream mechanisms related to spindle anchoring. This would be consistent with the GMC being a polarizing cue for larval brain NBs.

Finally, we tested whether GMC-dependent division axis maintenance acts in the same pathway as the previously reported

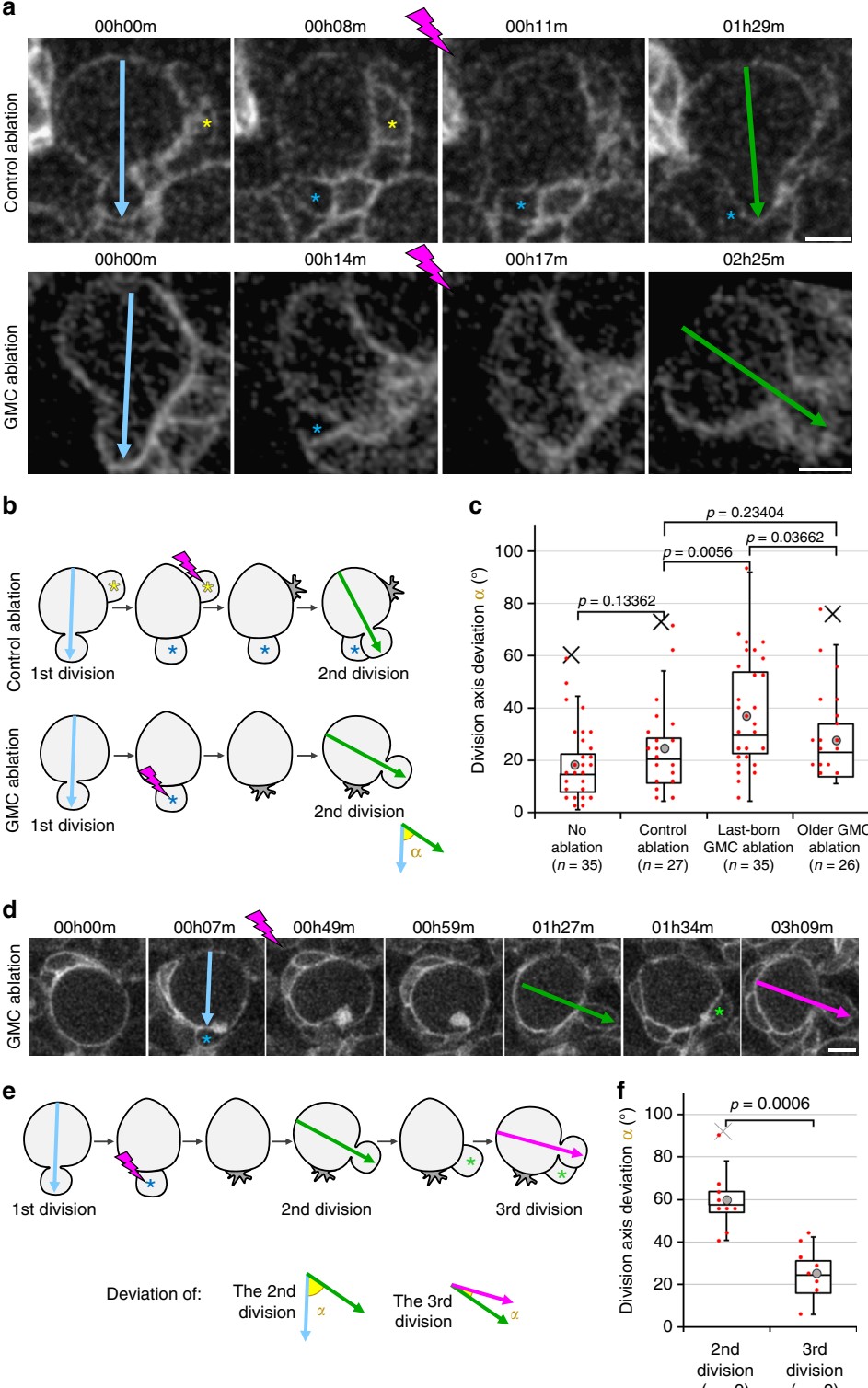

**Fig. 2** GMC ablation disrupts division axis maintenance. **a** Two successive divisions of NBs expressing *worniu*-GAL4-driven PH::GFP, the second division following the control ablation (upper panels) of a NB-neighbouring cell (yellow asterisk) away from the last-born GMC (blue asterisk) or following GMC ablation (lower panels) are shown. The ablation takes place between the two frames separated by a lightning symbol. Arrows: division axis. **b** Schematic of the ablations and angle measurements. **c** Deviation of the division axis following no ablation (Average angle: 18 ± 14°, *n* = 35), control ablation (23 ± 16°, *n* = 27), last-born GMC ablation (36 ± 20°, *n* = 35; also shown in Figs. 3c and 5c) or older GMC ablation (27 ± 16°, *n* = 26). Data from ten independent experiments. For examples of control or last-born GMC ablations resulting in a high misalignment of the second division axis, see Supplementary Movies 6 and 7. **d** Three successive divisions of NBs expressing *worniu*-GAL4-driven PH::GFP. The ablation takes place between the two frames separated by a lightning symbol. Blue asterisk: GMC prior to ablation, green asterisk: new last-born GMC. Arrows: division axis. **e** Schematic of the angles α measured from the movies shown in the previous panel. **f** Deviation α of the second (59 ± 14°, *n* = 9) and third (25 ± 11, *n* = 9) divisions described in the two previous panels. Data from five independent experiments. Scale bar in all panels: 5 μm

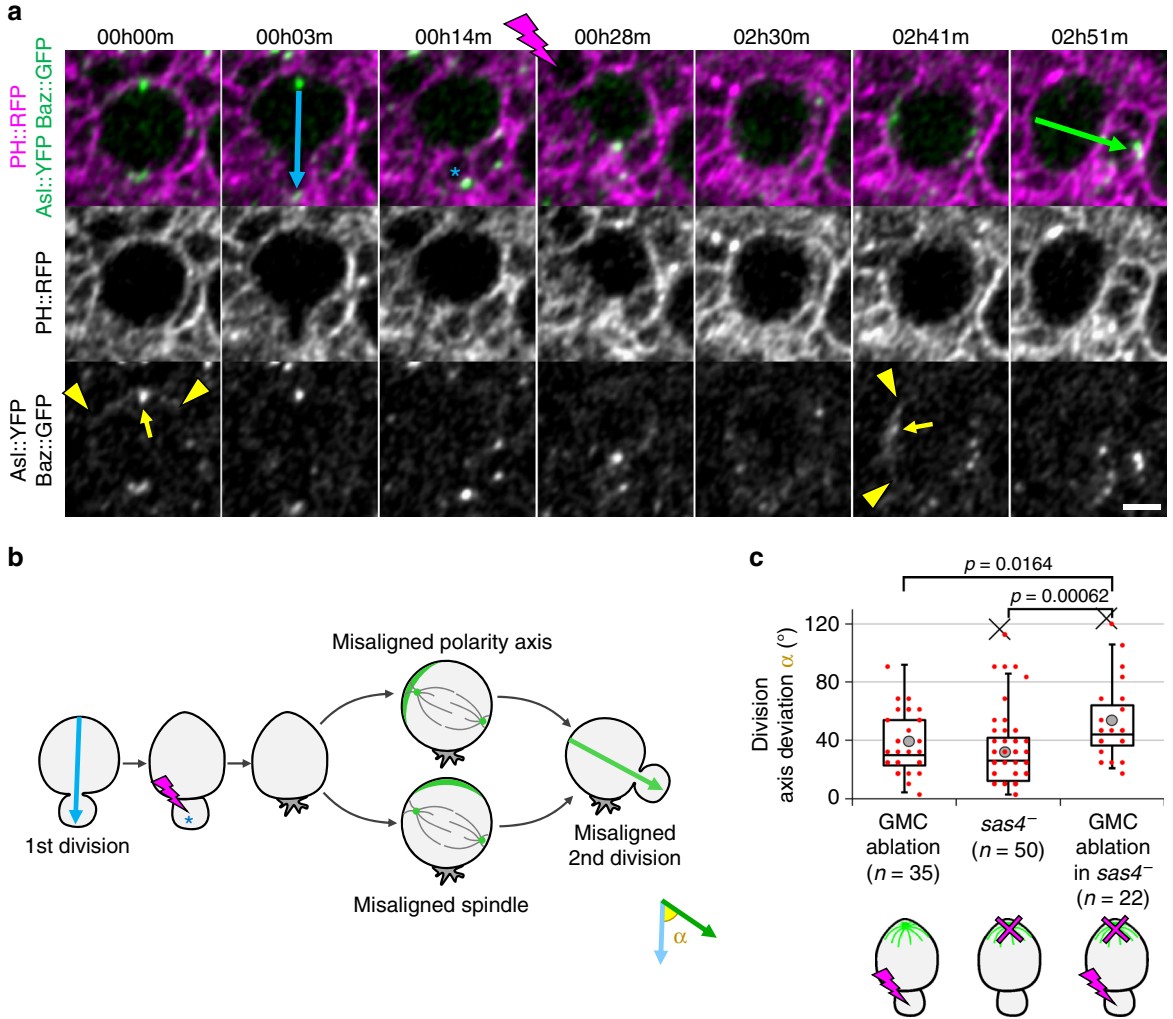

**Fig. 3** Last-born GMC ablation displaces NB polarization. **a** Two successive divisions of NBs expressing the membrane marker PH::RFP (magenta), the apical marker Baz::GFP and the centrosome marker Asl::YFP (green) pre- and post-ablation. The ablation takes place between the two frames separated by a lightning symbol. Blue and green arrows: division axis. Yellow arrows: apical centrosome. Arrowheads: limits of Baz::GFP apical crescents. Blue asterisk: GMC prior to ablation. Data from two independent experiments. Scale bar: 5 μm. **b** Schematic of the two possibilities explaining misaligned divisions following GMC ablation. Green crescent: apical pole. Grey lines: mitotic spindle. Green dots: centrosomes. **c** Deviation of the division axis following GMC ablation in control NBs (36 ± 20°, $n = 35$, also shown in Figs. 2c and 5c, data from ten independent experiments), no ablation in *sas4* mutant NBs (33 ± 26°, $n = 50$, also shown in Supplementary Fig. 1d) and GMC ablation in *sas4* mutant NBs (49 ± 28°, $n = 22$), data from six different experiments

intrinsic polarity cue for larval NBs[32]. We reasoned that, in this case, the division axis maintenance defect upon disruption of the intrinsic polarity cue division axis in *sas4* mutants should not increase upon ablation of the last-born GMC. On the contrary, the division axis deviation of *sas4* mutant NBs increased significantly upon GMC ablation (Fig. 3c). This supports the possibility that the GMC is an extrinsic polarizing cue contributing to NB division axis maintenance in parallel to the intrinsic, *sas4*-dependent cues.

**The NB/last-born GMC interface has specific features**. We next sought to understand the molecular mechanism for the ability of the GMC to affect NB division orientation. We hypothesized that the interface between the NB and its latest daughter cell was likely to mediate this function and that this interface might have specific characteristics distinguishing it from contacts between the NB and older daughter cells. Certain modes of divisions in budding yeast are oriented by a "division scar"[18] as a landmark guiding the orientation of the next division. We hypothesized that NBs could use a similar strategy, involving the midbody as a

polarizing cue, and tracing the midbody marker Pavarotti-GFP[35], we found that the midbody was present at the newly formed NB/GMC interface from cytokinesis onward and that the midbody from the previous division was inherited by the GMC (Fig. 4a, 34/51 cell cycles). Thus in most cases the NB cortex harbours one midbody marking the position of the last-born GMC. However, tracing the fate of the midbody over several divisions in multiple NBs revealed that in some cases the midbody was internalized by the NB during interphase (Supplementary Fig. 3a, 17/51 cell cycles). Comparing the deviations in division orientation between NBs that internalized the midbody and those that did not revealed no significant difference (Supplementary Fig. 3b). Therefore, the midbody itself is unlikely to serve as a landmark directly read by the NB to maintain its division orientation.

However, we noticed other specific features of the NB/GMC interface in the vicinity of the midbody. NBs expressing membrane markers such as the GTPase Rap1[36] or the PI(4,5) P2-specific PH domain of phospholipase Cδ fused to GFP (PH::GFP, {Claret:2014ig}) (Supplementary Movie 8) displayed structures resembling long tubules largely restricted to and

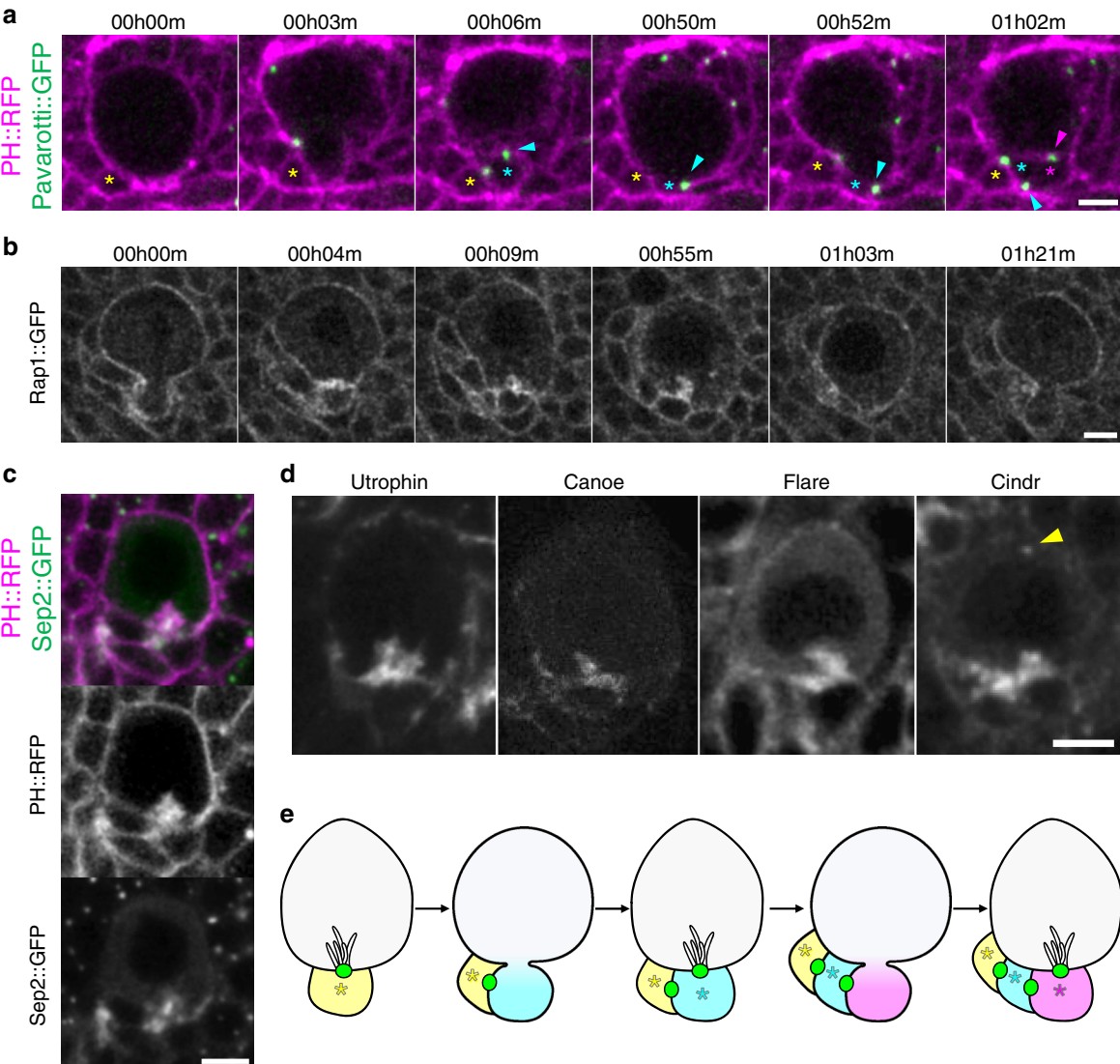

**Fig. 4** The NB/GMC interface presents specific characteristics. **a** Two successive divisions of NBs expressing the membrane marker PH::RFP (magenta) and the midbody marker Pavarotti::GFP (green). Yellow asterisk: previously born GMC, cyan asterisk: GMC formed at subsequent division, magenta asterisks: last-born GMC in this series. Blue arrowhead: midbody formed at the first division recorded stays at the newly formed NB/GMC interface but is inherited by the GMC in the next division, when a new midbody remains at the subsequently formed NB/GMC interphase (magenta arrowhead). **b** NB expressing the membrane marker Rap1::GFP. Tubular plasma membrane extensions originating from the NB/GMC interface appear within ~ 5 min following closure of the cytokinesis ring (00h04m–00h09m), persist throughout interphase (00h55m) and disappear upon cell rounding at the onset of mitosis (01h03m). **c** Interphase NB expressing Sep2::GFP (green) and the membrane marker PH::RFP (magenta). **d** Interphase NBs expressing GFP or YFP-fusions to the indicated proteins localizing to similar tubular extensions at the NB/last-born GMC interface. Cindr::GFP also localizes to centrosomes (arrowhead). **e** Schematic illustrating the specific characteristics of the NB/last-born GMC interface and the inheritance of this interface by the GMC at each division. Green: midbody. Asterisk: GMCs, same colour code as Fig. 3a. Scale bar in all panels: 5 μm

expanding from the NB/GMC interface into the NB cytoplasm (Fig. 4b). They systematically formed around the Septin2::GFP-labelled midbody (Fig. 4c) within the 5 min following closure of the cytokinetic furrow and were maintained throughout interphase, until they disappeared when NBs entered prophase (Fig. 4b). Consistent with the possibility that tubules may participate to orienting NBs, they were maintained at the NB/GMC interface in the cases when GMCs migrated away from their birthplace (Fig. 1a, 01h44m) and the cases when the midbody was internalized (Supplementary Fig. 3a, 01h01m). They were, however, also maintained following ablation of the last-born GMC (Fig. 2b, 00h49m, 00h59m).

Further examination of the NB/GMC interface revealed that it was rich in F-Actin (Fig. 4d). This prompted us to examine the

subcellular localization of several actin regulators using endogenously expressed fluorescent protein traps. We found that Flare (actin depolymerizing factor,[37]), Canoe (Afadin,[38]) and Cindr[39] (Supplementary Movie 9) were present at the NB/GMC interface (Fig. 4d).

We further reasoned that adhesion molecules could be involved in division orientation maintenance and might be enriched at the NB/GMC interface. We focussed on the adhesion molecule E-Cad, as it is involved in orienting mitosis in the fly sensory organ precursor lineage[13], is expressed in NBs and their lineage and has been reported to be enriched between NBs and their daughter cells[40,41]. However, neither E-Cad nor its binding partner β-Catenin were restricted to the interface of NBs and the last-born GMC compared to interfaces between NBs and older

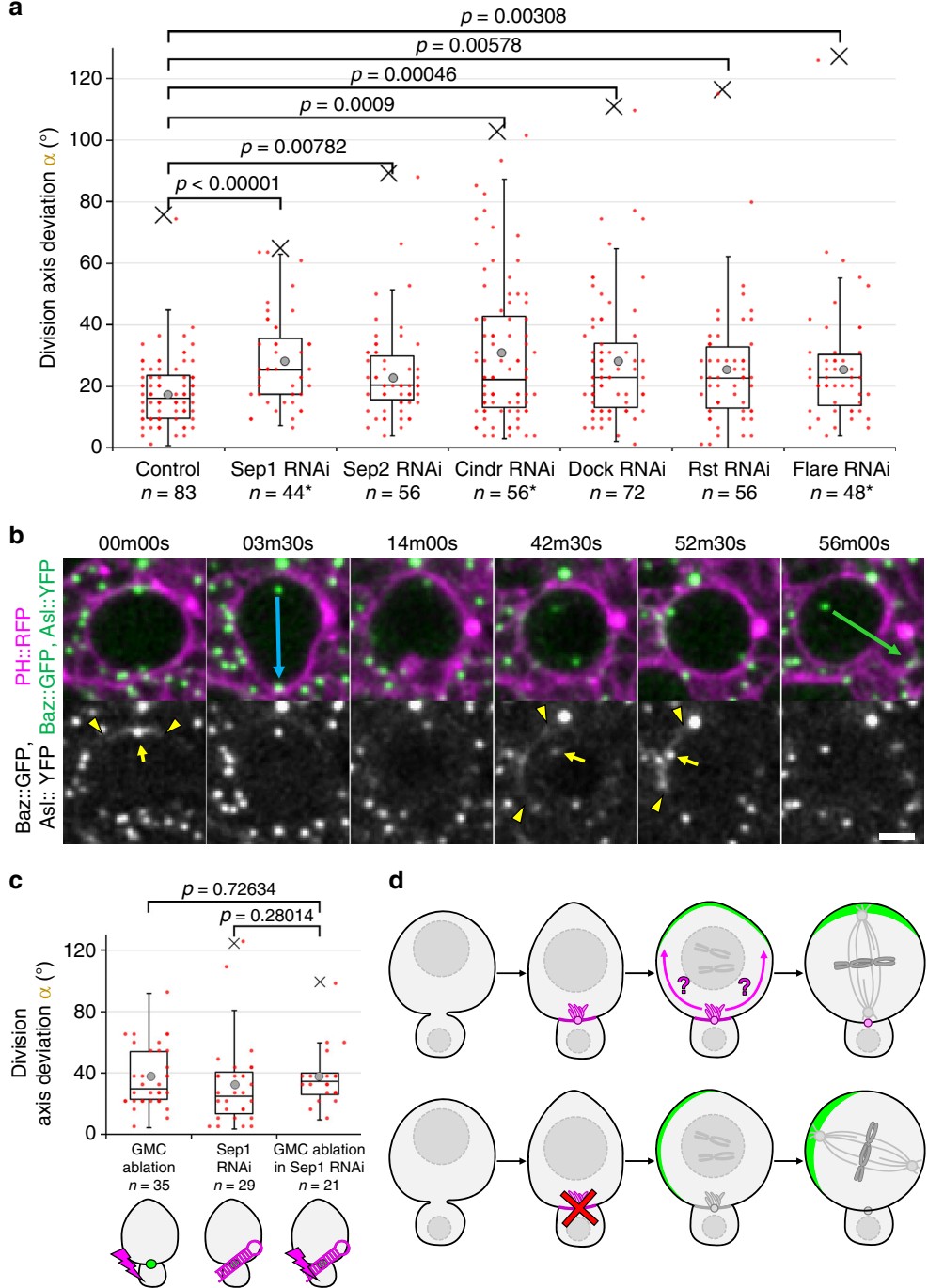

**Fig. 5** Components of the NB/GMC interface contribute to NB division axis maintenance. **a** Deviation of the division axis from one cell cycle to the next in NBs expressing *worniu*-Gal4-driven RNAi against the indicated components of the NB/GMC interface, as well as the Cindr interactors Rst and Dock. Asterisks: candidates for which protein depletion was confirmed. See Methods for precise genotypes. Data from, respectively, 2, 2, 1, 2, 2, 1 and 3 independent experiments. **b** Two successive division of a NB expressing *worniu*-Gal4-driven Cindr RNAi. The second division axis (green arrow) deviates from the first one (blue arrow) by 53° (quantified in **a**), due to the mispositioning of the apical Baz crescent (delimited by arrowheads), whereas the apical centrosome (arrow) aligns properly with this crescent. Proper spindle alignment was always observed (n = 5/5, 4/4, 17/17, 12/12, 4/4 and 6/6 cases with deviation >45°, respectively, in Sep1, Sep2, Cindr, dock, Rst and Flare RNAi). Scale bar: 5 μm. **c** Deviation of the division axis following GMC ablation in control NBs (36 ± 20°, n = 35, also shown in Figs. 2c and 3c, data from ten independent experiment), in NBs expressing *worniu*-Gal4-driven Sep1 RNAi (32 ± 27°, n = 29) and following GMC ablation in NBs expressing *worniu*-Gal4-driven Sep1 RNAi (37 ± 19°, n = 21). Data from three independent experiments. **d** Graphical summary of the findings. Top row: the NB/last-born GMC interface, distinguishable from other interfaces by the presence of the midbody and actin-rich plasma membrane extensions (magenta), biases through an unknown mechanism (magenta arrows) the apical polarization (green crescent) of NBs as they round up in mitosis. The mitotic spindle aligns with this crescent, resulting in division axis maintenance. Bottom row: disruption of this mechanism leads to misplaced apical polarization (with which the mitotic spindle still properly aligns) resulting in defective division axis maintenance

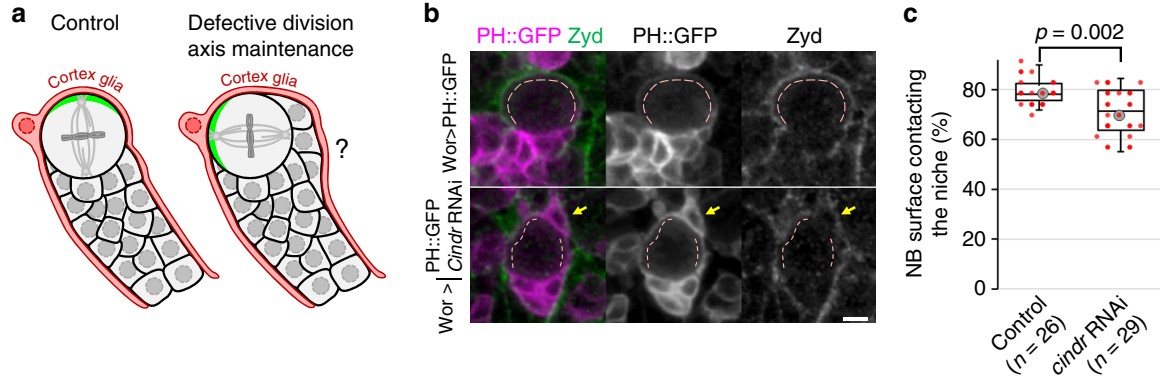

**Fig. 6** Reduced NB/glial cell contact area in *cindr* RNAi NBs. **a** Schematic of how defective division axis maintenance might alter NB/cortex glia contact area. **b** Control and *cindr* RNAi-expressing NBs expressing Wor-GAL-driven PH::GFP and stained for the cortex glia membrane marker Zyd. Dashed lines: NB/cortex glia contact. Arrow: a cell belonging to the NB progeny (as it is ensheathed with it by the Zyd-positive glia) but positioned away from other GMCs produced by the same NBs. Scale bar: 5 µm. **c** Percentage of the total NB surface in direct contact with the cortex glia cell in control and *cindr* RNAi-expressing NBs. Data from two independent experiments

GMCs (Supplementary Fig. 4a-b). In conclusion, specific characteristics distinguish the interface between the NB and its last-born daughter cell: the presence of a midbody, plasma membrane extensions, and an accumulation of actin and actin regulators.

**Molecules involved in NB division orientation maintenance**. We next used NB and NB progeny-specific RNAi to test whether components of the midbody and the NB/GMC interface are involved in orienting the axis of NB divisions. Efficient depletion of E-Cad by RNAi (Supplementary Fig. 4b) did not result in significant division orientation defects of central brain NBs (Supplementary Fig. 4c). Therefore, despite being involved in controlling niche position of mushroom body NBs[42], E-Cad is not critical for NB division orientation maintenance in the central larval brain.

Surprisingly, although midbody internalization had no effect on division axis maintenance (Supplementary Fig. 3), efficient depletion of the midbody components Septin 1 (Supplementary Fig. 4d-e) as well as RNAi against Septin 2 led to significant division orientation defects (Fig. 5a). Septins are required for cytokinesis in certain tissues[43] raising the possibility that cytokinesis failure indirectly disrupts division axis orientation. However, multinucleation in Septin1- and Septin2-depleted NBs was never observed (Supplementary Fig. 4). This is either because depletion of Septins was only partial (Supplementary Fig. 4d, e) or because Septins are not required for cytokinesis in NBs, as it is the case in other contexts[44,45]. Furthermore, efficient depletion of Flare and Cindr (Supplementary Fig. 4f, g) resulted in a significant increase of the division axis deviations (Fig. 5a). This prompted to express RNAi against a Cindr interactor, the transmembrane immunoglobulin Roughest (Rst,[46]), and against its adaptor Dreadlock (Dock[47]), depletion of all of which also affected division axis maintenance (Fig. 5a).

Like misaligned divisions following ablation of the GMC, misaligned divisions caused by RNAi against Flare, Cindr, Rst and Dock were associated with misplaced apical crescents rather than spindle–cortical polarity alignment problems (Fig. 5b, Supplementary Movie 10). We further reasoned that misplaced apical crescents, instead of being caused by the disruption of an additional polarizing cue relying on the NB/GMC interface, could be caused by an abnormally mobile apical centrosome directing polarization at the wrong place (Supplementary Fig. 4h). However, although we did occasionally observe apical centrosome detachment during interphase, this occurred at the same

frequency in control and RNAi-expressing NBs (control: 4/68; Sep1 RNAi: 4/47; Sep2 RNAi: 2/53; Cindr RNAi: 6/86; Dock RNAi: 7/73; Rst RNAi: 2/53; Flare RNAi: 6/55). Furthermore, precise measurements of the apical centrosome position suggest that, at least in Flare-depleted NBs, the apical centrosome position when NBs polarize does not correspond to the apical crescent position in metaphase (Supplementary Fig. 4h-j). Therefore, misplaced Baz crescents upon RNAi depletion of these components are unlikely to be caused by an incorrectly positioned apical centrosome.

Finally, we hypothesized that, if depleting components of the NB/GMC interface disrupts the role of the GMC as a polarizing cue, GMC ablation together with depleting these components should not further increase the resulting deviations of the orientation of NB division. Consistent with this idea, GMC ablation in Sep1-depleted NBs did not significantly increase division axis deviations when compared to Sep1-depleted NB alone or control NBs on which GMC ablations were performed (Fig. 5c). Based on these results, we propose that the role of the GMC in maintaining the division axis of NBs is mediated by proteins specifically localizing to the newly formed interface between the NB and its last-born daughter cell (Fig. 5d).

**Division orientation changes reduce NB/glia contact area**. In the larval brain, NBs and their progeny are ensheathed by glial cells[40] that provide proliferative signals[48] and protection against starvation[49] and oxidative stress[50]. We reasoned that, by placing daughter cells between the NB and their glial cell niche, inaccurate maintenance of division orientation could isolate the NBs from glial cells (Fig. 6a). Indeed, we measured that a significantly smaller portion of the NB surface contacted the ensheathing cortex glial cell in Cindr-depleted NBs (71 ± 9%) compared to controls (79 ± 5%, Fig. 6b, c). We further examined whether the reduction of this contact area may affect the niche's ability to protect NBs against oxidative stress. Consistent with this possibility, we observed that the proliferation of Cindr-depleted NBs, but not of controls, was significantly affected by experimentally induced oxidative stress (Supplementary Fig. 5A-F).

**Discussion**
Deciphering the signals that provide positional information is a central issue in understanding how cell divisions are oriented. Here we addressed this question in the highly proliferative NBs in the *Drosophila* larval brain, which maintain their division axis from one cell cycle to the next in part by using an apical

microtubule network as a spatial cue to specify their apico-basal polarity axis and consequently the orientation of mitosis[32]. We sought to understand why NBs only partially fail to maintain their division axis upon loss of this intrinsic polarizing cues and found out that the last-born daughter cell of NBs participates to their division axis maintenance. Our results also shed light on some aspects of the physiological importance of division axis maintenance in larval NBs, which has remained elusive. Control of NB division orientation may provide a means to maximize NB/cortex glia surface area to allow optimum protection against environmental stresses by the cortex glia[50]. Under normal conditions, about 80% of the surface of NBs is in direct contact with a cortex glia and NBs with partially defective division axis maintenance display reduced contact with cortex glia (Fig. 6). This most likely directly results from NBs producing progeny between themselves and the cortex glia when the last-born daughter cell derived cue that positions normally the apico-basal polarity axis is damaged. This seems to be important for the protective function of these glial cells on NB proliferation under stress conditions. Indeed, NBs with reduced surface contact to cortex glia appear to be less well protected by glial cells, as we observe a significant increase of sensitivity to oxidative stress (Supplementary Fig. 5) using an established assay[50]. However, despite this reduction being statistically significant, we measured only a 9% reduction in NB/cortex glia contact area. On a normal diet, addition of the oxidant tert-butyl hydroperoxide (tbh) results in a 14% drop in NB proliferation when the formation of lipid droplets mediating this protection is prevented[50]. It is therefore surprising that in our experiments reducing the NB/cortex glia contact area by only ~9% in Cindr-depleted NB (Fig. 6) is already accompanied by a similar drop in proliferation upon tbh treatment (Supplementary Fig. 5). Therefore, although this decrease may directly result from interfering with the protection provided by cortex glia, we cannot rule out other unrelated functions of Cindr in protecting NBs against the effect of tbh.

We initially hypothesized that the last-born GMC could act as an additional, extrinsic cue maintaining NB division orientation. A number of our observations are consistent with this possibility: we observed that, upon (perhaps artefactual) last-born GMC movements, NBs realign their division axis toward this GMC (Fig. 1); ablation of the last-born GMC (Figs. 2 and 3) and depletion of proteins specifically observed at the last-born GMC/NB interface (Fig. 5) affect division axis maintenance by misorienting the apico-basal polarity of NBs. We cannot exclude that the entire NB and any intrinsic spatial cue that it carries simply rotate upon migration or ablation of the last-born GMC or depletion of proteins specifically observed at the last-born GMC/NB interface. Thus the last-born GMC may participate in division axis maintenance by preventing NB rotation. This function could be mediated by specific adhesive contacts at the interface with the NB, plausible given the numerous specific characteristics that we have observed at that interface (Fig. 4). In particular, the midbody carried by this interface, although not likely to act itself as a stable physical link given its possible ability to migrate within the fluid mosaic of the plasma membrane[51] and the fact that its internalization does not affect division orientation maintenance (Supplementary Fig. 3), may be able to organize specific adhesive contacts at the NB/last-born GMC interface.

An alternative hypothesis is that the last-born GMC provides a cue that more directly functions in specifying the orientation of the apico-basal polarity axis by polarizing Baz, which functions upstream of NB division orientation control. Consistently, despite affecting division orientation maintenance, neither GMC ablation nor RNAi of Cindr disrupt alignment of the mitotic spindle with the polarity axis (Figs. 3a and 5b). In this case, the molecular mechanism through which a positional information provided by

the last-born GMC is transduced to the NB polarization machinery remains to be determined. Although bearing similarities with division axis maintenance in budding yeasts, relying on a Septin-rich cytokinesis remnant[18], the midbody of NBs is unlikely to directly control polarization as midbody internalization does not affect division axis maintenance (Supplementary Fig. 3). Instead, we propose that the midbody may organize various other specific components of the last-born GMC/NB interface that in turn may directly control NB polarization. This could be the case of cell–cell contacts organized by the midbody, consistent with the involvement of an adhesion molecule such as Roughest (Fig. 5a), whose mammalian orthologue physically interacts with Septins[52], and the fact that GMC ablation, although not directly targeting the interface, affects division axis maintenance. Another promising candidate potentially controlling NB polarity are the plasma membrane tubules probably organized by the midbody, given their physical origin (the midbody) and the timing (immediately after cytokinesis) of their appearance (Fig. 4, Supplementary Movie 8). Interestingly, a physical interaction was observed between Septins and the mammalian orthologue of Cindr[52], found enriched at the tubules (Fig. 4) and involved in division axis maintenance (Fig. 5). Tubules function might be linked to the integrity of the last-born GMC/NB interface, which itself probably depends on the integrity of the last-born GMC. While these tubules do not disappear upon GMC ablation (Fig. 2d), it would be of particular interest to monitor whether tubules morphology, dynamics or the enrichment of Flare and Cindr are affected by ablation of the last-born daughter cell.

Interestingly, proteins that we found involved in division axis maintenance were described to interact with polarity complexes in other contexts: Septins genetically interact with Baz during *Drosophila* embryogenesis[53], and the mammalian orthologues of Roughest regulate podocyte polarity by physically interacting with Par-3[54]. However, both Septins and Roughest localize to the basal pole of NBs, whereas Baz polarizes apically. Therefore, how could a cue received at the basal pole direct polarization of Baz, at the opposite apical pole of the NB? In the *C. elegans* zygote, the sperm entry point acts as a cue inducing an actomyosin flow[16] establishing Par complex polarity at the opposite end of the cell (see ref. 55 for a review). Septins[56], Cindr, Roughest[39] and Flare[37] can be linked in one way or another to the regulation of actomyosin, and at least the maintenance of Baz localization in mitotic NBs is also actin-dependent[57]. Intracellular long-range control of polarization has been further observed in eight-cell stage mouse blastomeres, where cell–cell contacts induce apical polarization at the opposite end of the cell[58]. A promising lead for future work is the possible involvement of actomyosin-dependent mechanical forces in such long-range control of polarity in NBs. Indeed, tensions participate in polarization in the *C. elegans* zygote[59,60], were proposed to mediate polarization of eight-cell stage mouse blastomeres[61] and maintain polarity in migrating neutrophils[62].

## Methods

**Fly stocks and genetics**. Flies were reared on standard corn meal food at 25 °C, except for RNAi-expressing larvae and their corresponding controls (Fig. 5 Supplementary Fig. 4), which were placed at 30 °C from the L1 larval stage to the L3 stage, at which point they were dissected. As RNAi was driven using the Worniu-GAL4 driver, which is not expressed in every NBs, UAS-nls::BFP was used as a GAL4 reporter to identify and exclude from the analysis NBs not expressing GAL4. For the genotypes of the animals used in each experiment, see Supplementary Table 1. For the origins of the stocks used, see Supplementary Table 2.

**Live imaging**. Every reference to Schneider's medium corresponds to glucose-supplemented (1 g l$^{-1}$) Schneider's medium (SLS-04–351Q). Live imaging was performed as described[63]. Entire brains were dissected from early L3 larvae (still crawling inside the food) in Schneider's medium and isolated from the surrounding imaginal discs. Particular care was taken to avoid pulling on brains at any time during the dissection and damaged brains were discarded. Isolated brains were

transferred to a drop of fibrinogen dissolved in Schneider's medium (50 mg ml$^{-1}$) on a 25 mm Glass bottom dish (WPI), which was then clotted by addition of thrombin (100 U ml$^{-1}$, Sigma T7513). Clots were then covered in Schneider's medium (approximately 750 μl spread over the entire surface of the glass).

RNAi-expressing brains and their associated controls (Fig. 5, Supplementary Fig. 4) were then imaged on a LEICA SP8 confocal microscope (LEICA) equipped with a ×63 NA 1.2 water immersion objective lens. Stacks of 25–30 optical z-section separated by 0.8 μm, covering a $132 \times 132 \times 20$–24 μm$^3$ region of the surface of the antero-ventral central brain were acquired every 210 s for 2 h 30 min to image Asl::YFP, Baz::GFP and PH::RFP, after which a final stack also imaged the GAL4 reporter Nls::BFP.

For laser ablations (Figs. 2, 3, 5c, Supplementary Fig. 2), brains were imaged on a Zeiss 710 confocal microscope equipped with a ×63 oil immersion objective lens. Stacks of 16 optical z-section separated by 1.2 μm, covering a $75 \times 75 \times 18$ μm$^3$ region of the surface of the antero-ventral central brain were acquired every 210 s for 2.5–3 h.

**Laser ablation**. Laser ablations were performed on a 710 confocal microscope (Zeiss) equipped with a ×63 oil immersion objective lens and a two-photon tunable Chameleon from Coherent set to 800 nm, using the fluorescence recovery after photobleaching module of the Zen software. Settings were as follows: laser intensity 25–32% (empirically adjusted depending on the depth of the targeted area within the tissue; targeted area $1.4 \times 1.4$ μm$^2$; 15 iterations.

**Image processing and angle measurement**. Data were processed and analysed using ImageJ[64]. A $0.8 \times 0.8 \times 0.8$ pixel-wide 3D Gaussian blur was applied to every image. For better visualization, a 0.75 gamma filter was applied to the pictures displayed in Fig. 4 and the associated movies.

**Angle measurement**. The 3D vectors corresponding to the division axis were defined by the 3D coordinates of the apical and basal centrosome at metaphase when a centrosome marker was available. When only a membrane marker was available, the 3D vectors were defined by the positions of the apical and the basal pole at telophase in three steps detailed in Supplementary Fig. 1a: (1) a manually determined axis bisecting the NB along its long axis is used to orthogonally slice a 3D stack covering the entire NB volume; (2) this orthogonal slice is used to determine the z coordinates of the apical and basal poles; (3) these z coordinates are used on the corresponding slices of the 3D stack to determine the x and y coordinates of the basal and apical poles.

The angle (α) between two 3D vectors was calculated using the formula:

$$\alpha = \arccos\left(\frac{\overrightarrow{A_1B_1} \cdot \overrightarrow{A_2B_2}}{\left|\overrightarrow{A_1B_1}\right|\left|\overrightarrow{A_2B_2}\right|}\right) \quad (1)$$

where the dot product is:

$$\overrightarrow{A_1B_1} \cdot \overrightarrow{A_2B_2} = \left(\left(x_{B_1} - x_{A_1}\right) + \left(y_{B_1} - y_{A_1}\right) + \left(z_{B_1} - z_{A_1}\right)\right) \times \left(\left(x_{B_2} - x_{A_2}\right) + \left(y_{B_2} - y_{A_2}\right) + \left(z_{B_2} - z_{A_2}\right)\right) \quad (2)$$

and the magnitude of any vector is:

$$\left|\overrightarrow{AB}\right| = \sqrt{(x_B - x_A)^2 + (y_B - y_A)^2 + (z_B - z_A)^2} \quad (3)$$

$x_{A_1}$ being for example the x coordinate of the apical centrosome during the first division and $y_{B_2}$ being for example the y coordinate of the basal centrosome during the second division.

**NB/cortex glia contact measurements**. High-resolution confocal stacks of PH::GFP-expressing, Zyd-immunostained NBs were acquired. The entire surface of NBs and the associated cortex glia were manually segmented, based, respectively, on the PH::GFP and the Zyd signals. The corresponding surfaces were then measured using the Isosurface function of the ImageJ BoneJ plugin[65].

**Proliferation assay**. Eggs were laid for 1 h in fly cages containing standard medium and yeast paste, in which larvae developed until 71 h after egg laying, after which they were transferred for 21 h to a 50% phosphate-buffered saline (PBS), 50% standard food mixture, supplemented with 0.2 mM 5-ethynyl-2′-deoxyuridine (EdU) and 0 or 15 mM tbh (Sigma). Their brains were then dissected and fixed for 20 min in 4% formaldehyde (Sigma), permeabilized in PBS–Triton 0.1% (PBT) overnight at 4 °C, rinsed once in PBS 0.5% bovine serum albumin (BSA), incubated 45 min in a Click-iT EdU Alexa Fluor 647 reaction cocktail (Thermo Fisher), rinsed once in PBS 0.5% BSA, stained for 4,6-diamidino-2-phenylindole, washed 3 times in PBS for 10 min, transferred to a 50% glycerol solution and mounted in a Vectashield mounting medium (Vector Laboratories). Mosaic tiles covering the entire volume of the brains were acquired on a LEICA SP8 confocal microscope, using

linear z-compensation to keep a high signal-to-noise ratio deeper in the tissue. The EdU signal being highly heterogenous between different brains and between different cells of the same brains, we trained the Pixel Classification workflow of Ilastik[66] to segment the EdU signal on small 3D sub-regions of various brains and then processed our entire 3D data using this training to generate signal probability maps, which were reliably segmented into binary masks by applying a 0.5 threshold. The resulting binary data were ultimately used to measure the volume of incorporated EdU, which was normalized to the precise time of exposure to EdU (between the transfer of larvae to EdU-containing medium to fixation, ranging from 20 to 21 h). See also (Supplementary Fig. 5).

**Statistical analysis**. P values displayed over boxplots were calculated using a nonparametric two-tailed Mann–Whitney U test. The numbers displayed in boxplots correspond to measurements in individual NBs. The number of animals used in each experimental data set is as follows: Fig. 1: 2. Fig. 2a–c: no ablation: 10; control ablation: 7; GMC ablation: 11. Fig. 2d–f: 4. Fig. 3a: 2. Fig. 3c: GMC ablation: 11; sas4$^-$: 7; GMC ablation in sas4$^-$: 8. Fig. 4a: 2. Fig. 4b: 3. Fig. 4c: 2. Fig. 4d Utrophin: 2; Canoe: 5; Flare: 6; Cindr: 6. Fig. 5a control: 4; Sep1 RNAi: 4; Sep2 RNAi: 2; Cindr RNAi: 3; Dock RNAi: 3; Rst RNAi: 2; Flare RNAi: 3. Fig. 5b: 3. Fig. 5c GMC ablation: 11; Sep1 RNAi: 4; GMC ablation in Sep1 RNAi: 4. Fig. 6b, c: control: 3; Cindr RNAi: 4; Supplementary Fig. 1 b-d: control: 10; sas4$^-$: 7. Supplementary Fig. 1e: 3. Supplementary Fig. 1f: 2. Supplementary Fig. 2 GMC ablation: 10; control ablation: 7. Supplementary Fig. 3a, b: 2. Supplementary Fig. 4a: 2. Supplementary Fig. 4b control: 2; E-Cad RNAi: 2. Supplementary Fig. 4c control: 4; E-Cad RNAi: 3. Supplementary Fig. 4d: 2. Supplementary Fig. 4e: 1. Supplementary Fig. 4f Cindr RNAi: 1; control: 1. Supplementary Fig. 4g Flare RNAi: 2; control: 3. Supplementary Fig. 4h control: 4; Flare RNAi: 3. Supplementary Fig. 5: control 0 mM tbh: 14; control 15 mM tbh: 14; Cindr RNAi 0 mM tbh: 10; Cindr RNAi 15 mM tbh: 12.

**Immunostainings**. For the β-Cat (Supplementary Fig. 3b) and Zyd (Fig. 6) immunostainings, brains were dissected in PBS, fixed for 20 min in 4% formaldehyde (Sigma), permeabilized in PBT for 1 h, incubated in a Rabbit-anti-β-Cat$^{central}$ antibody[67] diluted 1:200 in PBT or a Rabbit-anti-Zyd antibody[68] diluted 1:1000 in PBT for 2 h, rinsed 3 times in PBT, washed 3 times in PBT for 10 min, incubated in a secondary Donkey-anti-Rabbit antibody to Alexa 594 (Thermo Fisher) diluted 1:1000 in PBT for 1 h, rinsed 3 times in PBS, washed 3 times in PBS for 10 min, transferred to a 50% glycerol solution and mounted in a Vectashield mounting medium (Vector Laboratories). Every step was performed at room temperature.

## Data availability
The data sets generated during and/or analysed during the current study are available from the corresponding author on reasonable request.

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

## Acknowledgements

We would like to thank A. Guichet, N. Brown, J. Raff, C. Doe, C. Gonzalez, T. Lecuit, A. Wilde, D. Glover and T. Littleton for providing reagents; I. Näthke for critical reading and K. Mckay for help with the RNAi screen. We also thank the Bloomington stock center and the VDRC for providing fly lines and P. Appleton and S. Swift from the Center for advanced Scientific Technologies for technical support for microscopy. Work in J.J.'s laboratory is supported by a Sir Henry Dale fellowship from Wellcome and the Royal Society (100031Z/12/Z and 100031Z/12/A). The tissue imaging facility is supported by the grant WT101468 from Wellcome.

## Author contributions

N.L. and J.J. designed the study. N.L. designed and performed the individual experiments. N.L. and J.J. analysed and interpreted the data and wrote the manuscript. J.J. acquired funding.

## Additional information

**Competing interests:** The authors declare no competing interests.

