## [Peer Review File · Nature Communications]

Reviewers' comments:

Reviewer #1 (Remarks to the Author):

This is an interesting manuscript that nicely shows that the orientation of division of *Drosophila* Neuroblasts is directed by contact with their last born daughter. The mechanisms that orient neural progenitors and stem cells are of importance as many studies have shown that disrupting the orientation of neural progenitor divisions in other systems including mammalian embryo CNS can influence the fate of their daughters.

The evidence that division orientation is directed by the last born daughter is compelling, and although the authors do not determine how this mechanism works they do rule out some suspects (midbody, Ecadherin and centrosomes) and show the involvement of others (Septins and Actin regulators). They also show the disruption to division orientation is likely downstream of a disruption in orientation of apicobasal polarity rather than an inability to orient the spindle correctly.

The most disappointing aspect of the work is the authors do not show whether disrupting the orientation neuroblast divisions has any important consequences. They speculate briefly on this in the discussion, but in this system we simply don't know whether it is important or not. Despite this I think this work reveals a novel polarity orienting cue in a complex multicellular organism and opens the way to some interesting future work on mechanism and physiological importance.

Authors should provide some quantification for the data shown in Figure 5B. There are currently no n numbers for this.

Reviewer #2 (Remarks to the Author):

This manuscript addresses how the division orientation of *Drosophila* larval neuroblasts is controlled. Division orientation can be important for tissue structure and differentiation so this is a very important problem. The paper is based on the observation that the position of one of the daughter cells that results from the neuroblast division (GMC) is correlated with the subsequent division orientation. This phenomenon is especially apparent because sometimes the GMC moves a significant distance from its birth site (because of mechanical tissue constraints?). When this happens, the authors find that the next division orientation is more likely to be aligned with the GMC's final position than with the original division orientation (that gave rise to the GMC's original birth site). The authors identify a cellular structure ("tubules") at the cell interface that they hypothesize is responsible for division axis orientation. Overall this is interesting work that should be published assuming the following issues are addressed.

* a trivial model is that the GMC appears to control the division axis because mechanical constraints forced by the cluster of previously born GMCs rotate the division axis. The

authors have not adequately discounted this model.

- * the quantification of division axis deviation upon GMC ablation (2A'') shows large deviations, but the movie showing this effect in 2A only shows a small deviation. A movie should be included showing a division that has a much larger deviation. It would also be useful to see a control ablation movie for a large deviation division.

- * In ablation experiments, the authors appear to be ablating the GMC immediately after division. It would be informative to know if the division axis is affected if the GMC is allowed to move before ablation.

- * if the tubule structure is instructive for division orientation, it should move along with a moving GMC. Does it?

- * furthermore, is the tubule structure lost upon ablation?

minor points:

- * the methods don't describe how the division axis was measured when a centrosome marker wasn't available

- * the title of the paper is trying to be too clever. Readers would be better served by a more informative title.

Reviewer #3 (Remarks to the Author):

In this manuscript, the authors monitor the division axis plane of the larval neuroblast in wild-type, under several RNAi conditions and upon GMC daughter cell ablation. The core of this manuscript is the discovery of a complementary cue provided by the last-born daughter cell which instructs the orientation of the division plane axis of the next *Drosophila* Neuroblast division. The authors show that after ablation of the last-born GMC daughter cell, the polarity axis is impaired resulting in defective orientation of the division axis. This suggests a correlation between the presence of the GMC daughter cell and the correct alignment of the next division plane axis. In particular, the authors show that the interface between the last-born GMC daughter cell and the newly formed Neuroblast show an accumulation of membrane protrusions, Actin filaments and Actin regulators which are maintained throughout interphase until the onset of the next division.

These are very important new observations that uncover a new layer of regulation to establish oriented mitosis during asymmetric division. Therefore, this manuscript deserves publication in NatComm. There are a number of minor points that could be addressed by the authors:

- It would add clarity to the manuscript if, in Figure 1, the authors showed an image a neuroblast with no strong movement of the GMC daughter cell which reflect the data of the graph Fig 1C. Indeed, the picture Fig 1A reflects a 45° angle between the first division axis and the second division axis.

- Also, it is not clear in the manuscript (except Figure1) whether in all the graphs the alpha angle measured is the angle with the previous division axis or the last-born GMC.

How can the authors explain the difference of the measurement in Figure 1C (GMC n=52) and the no-ablation control in Figure 2 A'' ? (nicely clustered in Figure 1 and quite spread in Figure 2). Furthermore, in the method section, the angle used by the authors have been reported to be the dot product of the two vectors over their Euclidean length while it should be the arccos of the dot product of the two vectors over their Euclidean length. Please correct or explain.

- In page 18, the authors state that "at least in Flare-depleted neuroblasts, the apical centrosome position when neuroblasts polarize does not correspond to the apical crescent position in metaphase (Figure S4G)". However, this is not shown by the Figure S4G. It would good to add a picture supporting the statement.

- The authors do not explain in the methods why they do not have negative angle. Is it the absolute value of alpha?

- In the figure legend of figure 1, the alignment of the second division axis and the last born GMC number of replicates have been inversed.

- In the figure legend of figure 3B, the "GMC ablation" is the "GMC ablation control" $18 \pm 14^\circ$ n=35 which is not what is shown in the graph Fig3B "GMC ablation".

Response to reviewer's comments

We were first of all delighted to see the overall very positive response to our manuscript and thank all reviewers for their constructive criticism, which we feel as strongly improved the manuscript. Please find below our *point-by-point* reply to the specific points raised in *green*. All changes in the revised manuscript are also highlighted in green.

Response to Reviewers' comments:

Reviewer #1 (Remarks to the Author):

This is an interesting manuscript that nicely shows that the orientation of division of *Drosophila* Neuroblasts is directed by contact with their last born daughter. The mechanisms that orient neural progenitors and stem cells are of importance as many studies have shown that disrupting the orientation of neural progenitor divisions in other systems including mammalian embryo CNS can influence the fate of their daughters.

The evidence that division orientation is directed by the last born daughter is compelling, and although the authors do not determine how this mechanism works they do rule out some suspects (midbody, Ecadherin and centrosomes) and show the involvement of others (Septins and Actin regulators). They also show the disruption to division orientation is likely downstream of a disruption in orientation of apicobasal polarity rather than an inability to orient the spindle correctly.

The most disappointing aspect of the work is the authors do not show whether disrupting the orientation neuroblast divisions has any important consequences. They speculate briefly on this in the discussion, but in this system we simply don't know whether it is important or not. Despite this I think this work reveals a novel polarity orienting cue in a complex multicellular organism and opens the way to some interesting future work on mechanism and physiological importance.

We agree with this important criticism and are delighted to report that we have now made significant progress towards addressing it.

In particular, we designed experiments along the following rationale: It has been shown that upon extrinsic stresses such as starvation or oxidative stress most cells in the larvae cease dividing, except for neuroblasts that are somehow protected by the action of glial cells (Bailey, Koster, Guillermier, et al., 2015). We have put forward the hypothesis that

by keeping the axis of division stable, neuroblast might maximize the area with which they are in contact with the surrounding cortical glial cells to optimize their protective effect. In the revised version we test this hypothesis and include now new data strongly suggesting that this might be indeed the case.

*We first measured in 3D the neuroblast/glia cell contact surface in intact larval brains. Staining brains with an antibody against Zydeco, specific to the relevant glial cells (Melom & Littleton, 2013), we find that the neuroblast/glia cell surface area upon cindr RNAi, which affects neuroblast division orientation, is significantly reduced when compared to controls. This is now **new figure 6** in the revised manuscript.*

*We then reproduced an assay used in the Bailey et al paper, in which stress is experimentally induced in larvae by feeding them chemicals that activate reactive oxygen species. The effect of this treatment on neuroblast proliferation can then be monitored by measuring the amount of cell division in entire brains (i.e the central brain and ventral nerve cord) based on EdU staining (Bailey, Koster, Guillemer, et al., 2015). The assumption is that in the central brain and ventral nerve cord, proliferating neuroblasts account for the majority of cell divisions. These measurements are thus very sensitive as they integrate hundreds of neuroblast divisions for each sample point. If optimal neuroblast/glia cell contact is important to protect neuroblasts from experimentally induced stress, an effect on the overall proliferation in the tissue would be expected if contact with glial cells is reduced (a consequence of damaging the division orientation memory). Importantly, our measurements indeed indicate that neuroblasts that less well maintain their division orientation have an increased sensitivity to oxidative stress. These data are now provided as **new Supplementary Fig. 5**.*

Therefore, the physiological relevance of orienting neuroblast divisions might be to ensure optimal protection by glial cells during the development of the central nervous system of the fly, which is in strong support of our hypothesis.

We mention these experiments now in the abstract (page 1, line19-21).

The experiments are detailed in the main text (page 12-13, line 258-268) and discussed (page 14-15, line 303-313)

Authors should provide some quantification for the data shown in Figure 5B. There are currently no n numbers for this.

*We added in the figure legend that division axis deviation was quantified in panel **A**, and now present the quantification for each RNAi of how many cases of important*

division axis misalignment (arbitrarily set to deviation >45°) correspond to misaligned crescent rather than misaligned spindles.

This can be found on page 31, line 730-733 in the revised manuscript.

Reviewer #2 (Remarks to the Author):

This manuscript addresses how the division orientation of *Drosophila* larval neuroblasts is controlled. Division orientation can be important for tissue structure and differentiation, so this is a very important problem. The paper is based on the observation that the position of one of the daughter cells that results from the neuroblast division (GMC) is correlated with the subsequent division orientation. This phenomenon is especially apparent because sometimes the GMC moves a significant distance from its birth site (because of mechanical tissue constraints?). When this happens, the authors find that the next division orientation is more likely to be aligned with the GMC's final position than with the original division orientation (that gave rise to the GMC's original birth site). The authors identify a cellular structure ("tubules") at the cell interface that they hypothesize is responsible for division axis orientation. Overall this is interesting work that should be published assuming the following issues are addressed.

* a trivial model is that the GMC appears to control the division axis because mechanical constraints forced by the cluster of previously born GMCs rotate the division axis. The authors have not adequately discounted this model.

We agree that it would be important to test whether the entire GMC cluster produced by a neuroblast and contacting it contributes or whether the effect we find is more specifically linked to the last-born daughter cell. We now provide new data to clarify this point. We investigated the possible influence of older GMCs by performing another set of experiments: instead of destroying the last-born GMC, we destroyed the GMC generated one cell cycle earlier. This led to a small but non-significant increase of the division axis deviation compared to control cuts. It is noteworthy that, the older GMC is necessarily directly in contact with the last-born GMC and ablating this older GMC potentially affect the last-born GMC itself via the same problems we raise in this section, i.e. generation of cellular debris, direct damages by the laser and deformation of neighboring cells toward the ablated cell. This is in agreement that older GMC cuts have similar effects than control cuts.

*Nonetheless, ablation of the last-born GMC results in a significantly higher deviation than ablation of the older GMC. This seems to indicate that the last-born GMC, and not the older progeny, is involved in polarizing neuroblasts which would be consistent with the specific characteristics observed at the contact between neuroblasts and their last-born GMCs. We added this data to **Figure 2A''**.*

This can be found in the main text of the revised manuscript on page 7 line 133-143 and in the legend of figure 2 (page 29, line 689-690).

We actually do think that mechanical constraints might be key to position the apicobasal polarity axis by a specific cue provided by the last-born daughter cell. Proving this point, however, requires extensive experimental assessment, that we feel is beyond the scope of this manuscript and right now a focus of our current research. We speculate about a potential involvement of mechanical constraints at the end of the discussion (page 16, line 330-341).

* the quantification of division axis deviation upon GMC ablation (2A'') shows large deviations, but the movie showing this effect in 2A only shows a small deviation. A movie should be included showing a division that has a much larger deviation.

*We decided to highlight the average deviation that we observed for this type of ablation in the previous version. We now changed this panel. The "GMC ablation" row of **Figure 2A** now shows a case with a larger deviation (previously the average value of 36°, now 58°).*

This can be found in revised figure 2A.

It would also be useful to see a control ablation movie for a large deviation division.

*Here we provide an example for the reviewer of one control ablation resulting in a 71° deviation, corresponding to the maximum outlier of **Figure 2A'', Control ablation**. The 2D vectors displayed within the plane of imaging show a seemingly higher (>90°) deviation than this, but the actual measurement between the corresponding 3D vectors yields a deviation of 71°. We are uncertain, however, how to justify this data in the main text and how to discuss it. We are happy to include it if advised by the reviewer, but decided to not show the panels, as the outliers are covered in figure 2A''.*

* In ablation experiments, the authors appear to be ablating the GMC immediately

after division. It would be informative to know if the division axis is affected if the GMC is allowed to move before ablation.

We agree that this would be in principle an interesting experiment. However, the proposed experiment is particularly difficult to perform for the following reasons:

- *GMC movements as obvious as the one displayed **Figure 1A** (and staying within the plane of the imaging) are rare and would be particularly hard to follow on the microscope that we use to perform laser ablation (compare **Supplementary Fig 1A**, acquired on our LEICA SP8 equipped with hybrid detectors, to **Figure 2A**, the same fluorophore acquired on our Zeiss 710 microscope without hybrid/gaasp detectors) especially without the filtering we apply on these images post-acquisition. It is not possible to predict, when such an event would happen.*
- *As the reviewer pointed out, we perform GMC ablations relatively early (not immediately, as ablation of the GMC within the 10 minutes following apparent closure of the cytokinetic furrow always results in the destruction of the neuroblast as well, probably because abscission has not taken place yet), i.e. 24.0 ± 7.4 minutes after telophase. This is because neuroblasts can enter in prophase anytime between 35 minutes and approximately 2 hours after the previous telophase, and we want to make sure that we perform the ablation before neuroblasts enter prophase (at which point they start polarizing).*
- *When performing the ablation experiment presented in the current version, we only performed 4 or 5 ablation attempts (which does not mean 5 successful ablations) per optic lobe in order to avoid damaging the tissue and could manage imaging 2 to 3 different lobes at the same time. In our experience, we think that the additional difficulty of reliably tracking GMC positions during the imaging would prevent us from imaging multiple lobes at the same time.*

Thus, we think that observing a statistically meaningful number of cases where we are able to follow significant GMC movements and then manage to perform an efficient GMC ablation before neuroblasts polarize again would be exceedingly difficult. Finally, we are not quite sure what would be the interpretation of the alignment of the division axis with the previous one division axis being differently affected or not by allowing GMC movement, as we ultimately destroy this GMC before neuroblasts polarize. The data shown in figure1A is nonetheless consistent with a role for GMCs in polarizing neuroblasts justifying the laser ablation experiments performed subsequently.

* if the tubule structure is instructive for division orientation, it should move along with a moving GMC. Does it?

Indeed, it does (as well as the midbody). Prompted by the reviewer's comment, we now mention this in the text. This can be found on page 9, line 190-194 in the revised manuscript. The magenta arrow corresponding to the neuroblast/GMC axis directly

points to the tubules in **Figure 1A** (the arrow pointing to it is a coincidence, the tubules were not used as a landmark to define this axis). The tubules are not apparent at every timepoint in this case as they moved out of the displayed focal planes, but can be visualized at the neuroblast/GMC interface throughout GMC movements by displaying different slices of the neuroblast volume (yellow arrows in the bottom row). However, we have no evidence so far that the tubule structures are actually important. We emphasize this in the discussion of the revised manuscript. This can be found on page 14, line 296-301 in the revised manuscript.

* furthermore, is the tubule structure lost upon ablation?

*As far as we can tell, the tubules are never lost upon ablation. Our clearest example is the case displayed in **Figure 2B**, showing that the tubules are still present immediately after the ablation. We added a panel (**Figure 2B, 00h59m**) showing, like in control, that the tubules remain present until neuroblasts round up again up and now mention these points in the texts. We also now speculate about the possible contribution of tubules in the discussion. This can be found on page 14, line 296-301 in the revised manuscript.*

minor points:

* the methods don't describe how the division axis was measured when a centrosome marker wasn't available

*We now cite the steps detailed in **Supplementary Fig. 1A** in the methods section. This can be found on page 20, line 387-394 in the revised manuscript.*

* the title of the paper is trying to be too clever. Readers would be better served by a more informative title.

We have changed the title of the manuscript and hope that this title is more informative.

New Title:

“The last-born daughter cell orients the apicobasal polarity axis of Drosophila larval neuroblasts »

Reviewer #3 (Remarks to the Author):

In this manuscript, the authors monitor the division axis plane of the larval neuroblast in wild-type, under several RNAi conditions and upon GMC daughter cell ablation. The core of this manuscript is the discovery of a complementary cue provided by the last-born daughter cell which instructs the orientation of the division plane axis of the next Drosophila Neuroblast division. The authors show that after ablation of the last-born GMC daughter cell, the polarity axis is impaired resulting in defective orientation of the division axis. This suggests a correlation between the presence of the GMC daughter cell and the correct alignment of the next division plane axis. In particular, the authors show that the interface between the last-born GMC daughter cell and the newly formed Neuroblast show an accumulation of membrane protrusions, Actin filaments and Actin regulators which are maintained throughout interphase until the onset of the next division.

These are very important new observations that uncover a new layer of regulation to establish oriented mitosis during asymmetric division. Therefore, this manuscript deserves publication in NatComm. There are a number of minor points that could be addressed by the authors:

- It would add clarity to the manuscript if, in Figure 1, the authors showed an image a neuroblast with no strong movement of the GMC daughter cell which reflect the data of the graph Fig 1C. Indeed, the picture Fig 1A reflects a 45° angle between the first division axis and the second division axis.

*We have now added a case showing no GMC movement and proper maintenance of the division axis to **Figure 1A**.*

- Also, it is not clear in the manuscript (except Figure1) whether in all the graphs the alpha angle measured is the angle with the previous division axis or the last-born GMC.

To avoid this confusion, we renamed the alignment of the division axis with the last-born GMC as “angle β ” (which is the only case where this kind of measurement is performed) in figure 1, whereas the alignment of division axis with the previous division axis remains as “angle α ”. Additionally, we stated more clearly in the text that “division axis deviation” corresponds to this angle α at the beginning of the “ablations” part, and renamed the y axis of box plots presenting the distribution of this angle as “Division

axis deviation α (°)" in all figures. This can be found on page 5, line 97-105 in the main text and on page 6, line 119.

How can the authors explain the difference of the measurement in Figure 1C (GMC n=52) and the no-ablation control in Figure 2 A''? (nicely clustered in Figure 1 and quite spread in Figure 2).

*This difference stems from the possible confusion stated above: the measurements in Figure 1C (GMC n=52) correspond to the alignment between the division axis and the GMC (what we now name "angle β "), whereas the no-ablation control of **Figure 2A''** corresponds to the alignment between the division axis and the previous division axis (what we now name "angle α "). To avoid this confusion, in addition to renaming the y axis of every box plot showing this kind of measurement, we added at the bottom left corner of **Figure 2A'** a cartoon reminding that the angle α measures the alignment between two division axes.*

Furthermore, in the method section, the angle used by the authors have been reported to be the dot product of the two vectors over their Euclidean length while it should be the arccos of the dot product of the two vectors over their Euclidean length. Please correct or explain.

Indeed, the formula that was used to calculate angles is the arccos of the dot product of two vectors over their Euclidean length, we thank the reviewer for signalling this mistake. This has been changed in the method section, page 20, line 396.

- In page 18, the authors state that "at least in Flare-depleted neuroblasts, the apical centrosome position when neuroblasts polarize does not correspond to the apical crescent position in metaphase (Figure S4G)". However, this is not shown by the Figure S4G. It would good to add a picture supporting the statement.

*We added panel **Supplementary Fig. 4G''** illustrating the statement.*

- The authors do not explain in the methods why they do not have negative angle. Is it the absolute value of alpha?

First, mathematically, as the reviewer pointed out above, the method used to calculate angles uses the arccos function $\alpha = \arccos(x)$, for which the range of usual principal value of α is always between 0 and 180°, regardless of the value of x (https://en.wikipedia.org/wiki/Inverse_trigonometric_functions).

Second, intuitively, since what we measure is simply how well two 3D vectors align, without considering any plane of reference, we think there should be no reason for the angle between them to be negative. Negative values would be relevant if, for, example, there was a chronological relationship between two vectors (e.g. the angle between \vec{A} and \vec{B} indicates a directional rotation of +50°), which would define a plane AB in which,

for example, counter clockwise rotation is defined as positive. Within this bi-dimensional polar coordinate system, other vectors (e.g. \vec{C} in the following illustration) may define rotations with negative angles.

- In the figure legend of figure 1, the alignment of the second division axis and the last-born GMC number of replicates have been inverted.

We thank the reviewer for spotting this mistake., which we have now corrected.

- In the figure legend of figure 3B, the "GMC ablation" is the "GMC ablation control" $18 \pm 14^\circ$ $n=35$ which is not what is shown in the graph Fig3B "GMC ablation".

Again, we appreciate the thoroughness, with which the reviewer has analysed our manuscript. We have now corrected this mistake.

REVIEWERS' COMMENTS:

Reviewer #1 (Remarks to the Author):

My main criticism of the original submission was that the authors did not show what the consequences of disruption of the NB division orientation were. Is it important? The authors have now made a reasonable stab at testing one hypothesis - that the regulation of division orientation ensures a full covering of the NB by its protective glia, i.e. it stops daughters getting in between the NB and the glia. Experimental manipulation of division orientation through depletion of Cindr goes so far as supporting this hypothesis (the architecture of NB and its daughters in relation to the glia is certainly disrupted) but as the authors discuss they are not sure they understand all the consequences of Cindr depletion in addition to misregulation of division orientation. None the less I think this adds significantly to the manuscript and improves its discussion of the functional relevance.

Overall I think this is a good piece of work that adds significantly to our understanding of the repertoire of mechanisms that can control division orientation within progenitor/stem cell zones.

Reviewer #2 (Remarks to the Author):

The authors have discovered a "tubule" structure that appears to form from the midbody of neuroblast divisions and could be involved in specifying the polarity/division axis of the subsequent division. This is an important and interesting discovery, and the authors present sufficient data to warrant publishing this finding. The authors also find that during some divisions, perhaps because of mechanical constraints from adjacent cells (as the authors argue in their response), the budding GMC causes the cell to rotate, shifting the position of the tubular structure and the polarity axis. Given these observations, I find that some of the author's interpretations and presentation are confusing/flawed because they focus more on the likely indirect role of the budding daughter cell than the instructive role of the tubular structure, and this should be corrected before publication. For example:

* The title: "The last-born daughter cell orients the apicobasal polarity axis of *Drosophila* larval neuroblasts". Throughout the text the focus is on how the daughter cell controls the polarity axis. While these statements may be technically correct, I believe they are very misleading because the last born daughter cell may orient the polarity axis by rotating the cell when mechanical constraints from adjacent cells cause it to do so. This would be an indirect mechanism according to the mechanism described above, which the authors haven't adequately discounted (and to be clear I don't believe that they need to resolve this issue with data - simply that they shouldn't come down so strongly on one side).

* Line 283: "We find that larval NBs are not exclusively polarized by cell intrinsic mechanisms" The authors make several statements along these lines and I disagree that their data support them. If the daughter cell is simply changing the division axis by rotating the cell, it is incorrect to say that it is "polarizing" the neuroblast.

* Note that it appears the authors realized this problem because they added revisions to downplay the role of the tubules, such as line 300: "Whether they [the tubules] participate

to polarizing NBs remains unanswered, and they appeared unaffected by ablation of the last-born GMC". However, I do not believe the authors have conclusively demonstrated that the tubules are unaffected by the ablation event.

I'm not certain but I believe this problem could be addressed with minor revisions to the text.

Additionally, I requested that the authors include two movies relevant to Figure 2A" - one illustrating a large angle deviation for a control ablation (second column) and one illustrating a large angle deviation for a last born GMC ablation. I would really like to see these movies and I believe they would be useful to in-depth readers. The authors did not include these movies in the revision, strangely stating "We are uncertain, however, how to justify this data in the main text and how to discuss it." This seems like a fairly straightforward problem to overcome - perhaps the editor can provide a suggestion?

REPLY TO REVIEWERS' COMMENTS:

Please find below our response to the reviewer's comments in blue.

Reviewer #1 (Remarks to the Author):

My main criticism of the original submission was that the authors did not show what the consequences of disruption of the NB division orientation were. Is it important? The authors have now made a reasonable stab at testing one hypothesis - that the regulation of division orientation ensures a full covering of the NB by its protective glia, i.e. it stops daughters getting in between the NB and the glia. Experimental manipulation of division orientation through depletion of Cindr goes so way to supporting this hypothesis (the architecture of NB and its daughters in relation to the glia is certainly disrupted) but as the authors discuss they are not sure they understand all the consequences of Cindr depletion in addition to misregulation of division orientation. None the less I think this adds significantly to the manuscript and improves its discussion of the functional relevance.

Overall I think this is a good piece of work that adds significantly to our understanding of the repertoire of mechanisms that can control division orientation within progenitor/stem cell zones.

We would like to thank the reviewer for supporting our manuscript.

Reviewer #2 (Remarks to the Author):

The authors have discovered a "tubule" structure that appears to form from the midbody of neuroblast divisions and could be involved in specifying the polarity/division axis of the subsequent division. This is an important and interesting discovery, and the authors present sufficient data to warrant publishing this finding.

We also would like to thank this reviewer for supporting our work and the constructive criticism.

The authors also find that during some divisions, perhaps because of mechanical constraints from adjacent cells (as the authors argue in their response), the budding GMC causes the cell to rotate, shifting the position of the tubular structure and the polarity axis. Given these observations, I find that some of the author's interpretations and presentation are confusing/flawed because they focus more on the likely indirect role of the budding daughter cell than the instructive role of the tubular structure, and this should be corrected before publication.

For example:

* The title: "The last-born daughter cell orients the apicobasal polarity axis of *Drosophila* larval neuroblasts".

Throughout the text the focus is on how the daughter cell controls the polarity axis. While these statements may be technically correct, I believe they are very misleading because the last born daughter cell may orient the polarity axis by rotating the cell when mechanical constraints from adjacent cells cause it to do so. This would be an indirect mechanism according to the mechanism described above, which the authors haven't adequately discounted (and to be clear I don't believe that they need to resolve this issue with data - simply that they shouldn't come down so strongly on one side).

* Line 283: "We find that larval NBs are not exclusively polarized by cell intrinsic mechanisms" The authors make several statements along these lines and I disagree that their data support them. If the daughter cell is simply changing the division axis by rotating the cell, it is incorrect to say that it is "polarizing" the neuroblast.

* Note that it appears the authors realized this problem because they added revisions to downplay the role of the tubules, such as line 300:

"Whether they [the tubules] participate to polarizing NBs remains unanswered, and they appeared unaffected by ablation of the last-born GMC". However, I do not believe the authors have conclusively demonstrated that the tubules are unaffected by the ablation event.

We agree that this is a formal possibility, that we cannot rule out and have added this element to the discussion (see below). We understand that the moving tubules intuitively suggest that the NBs is rotating along with the migrating GMC. However, the tubules are likely to be midbody associated structures and midbodies have been shown in certain context to be able to move within the fluid mosaic of lipids of the plasma membrane (Daniel et al., 2018). Since the midbody belongs to the plasma membrane of both the neuroblast and the GMC, given this topology, it

would be hard to conceive how the midbody and its associated structures would not follow the migration of the GMC. Therefore, changes of the position of the tubules, do not necessarily indicate neuroblast rotations.

We now refer to the possibility that the GMC might normally prevent neuroblast rotation in the revised discussion (page 16, line 369 onwards) and no longer conclude that the GMC acts as a polarizing cue and have been more accurate throughout the results section to not overinterpret our results (see changes line 99, 104, 135, 192, 197, 216 and the headings 168 and 258).

We have further changed the title to be more accurate to make room for more several interpretations to: "The last-born daughter cell contributes to division orientation of *Drosophila* larval neuroblasts"

We also agree, that our previous statement that tubules "appeared unaffected by ablation of the last-born GMC" is misleading and could be interpreted as "we have conclusive evidence that the tubules are in no way affected by the ablation". It could be further extrapolated from this that we imply that tubules are unlikely to polarize NBs, which is not the message that we were trying to convey.

To be more accurate: tubules remain detectable after GMC ablation, but we cannot tell whether their morphology or dynamics and composition are affected following ablation, because of the low spatio-temporal resolution and the poor signal-to-noise ratio of the movies acquired on the microscope with which the ablations were performed. We now give more wait to a potential role of the tubules and discuss the possibility of a hierarchical relationship between the last-born GMC, its interface with the NB and the tubules (page 16, line 399 onwards).

I'm not certain but I believe this problem could be addressed with minor revisions to the text.

We thank the reviewer for this opportunity and have hopefully satisfactorily done as advised.

Additionally, I requested that the authors include two movies relevant to Figure 2A" - one illustrating a large angle deviation for a control ablation (second column) and one illustrating a large angle deviation for a last born GMC ablation. I would really like to see these movies and I believe they would be useful to in-depth readers. The authors did not include these movies in the revision, strangely stating "We are uncertain, however, how to justify this data in the main text and how to discuss it." This seems like a fairly straightforward problem to overcome - perhaps the editor can provide a suggestion?

We now added the requested movies as new supplementary movies 6 and 7 and refer to them in the relevant figure legends (page 29, line 930 onwards).